# Modal Identification of Ultralow-Frequency Flexible Structures Based on Digital Image Correlation Method

**Hui Qian** [1] , **Yimeng Wu** [1] , **Rui Zhu** [2] , **Dahai Zhang** [2] **and Dong Jiang** [1],*

[1] School of Mechanical and Electronic Engineering, Nanjing Forestry University, Nanjing 210037, China; qianhui@njfu.edu.cn (H.Q.); wuyimeng@njfu.edu.cn (Y.W.)

[2] Institute of Aerospace Machinery and Dynamics, Southeast University, Nanjing 211189, China; rui_zhuy@163.com (R.Z.); dzhang@seu.edu.cn (D.Z.)

* Correspondence: jiangdong@njfu.edu.cn

**Abstract:** Traditional modal testing has difficulty accurately identifying the ultralow-frequency modes of flexible structures. Ultralow-frequency excitation and vibration signal acquisition are two main obstacles. Aiming at ultralow-frequency modal identification of flexible structures, a modal testing method based on Digital Image Correlation method and Eigensystem Realization Algorithm is proposed. Considering impulse and shaker excitation are difficult to make generate ultralow-frequency vibration of structures, the initial displacement is applied to the structure for excitation. The ultralow-frequency accelerometer always has a large mass, which will change the dynamics performance of the flexible structure, so a structural vibration response was obtained through the Digital Image Correlation method. After collecting the free-decay vibration signal, the ultralow-frequency mode of the structure was identified by using the Eigensystem Realization Algorithm. Ground modal tests were conducted to verify the proposed method. Firstly, a solar wing structure was adopted, from which it was concluded that the signal acquisition using Digital Image Correlation method had high feasibility and accuracy. Secondly, an ultralow-frequency flexible cantilever beam structure which had the theoretical solution was employed to verify the proposed method and the theoretical fundamental frequency of the structure was 0.185 Hz. Results show that the Digital Image Correlation method can effectively measure the response signal of the ultralow-frequency flexible structure, and obtain the dynamics characteristics.

**Keywords:** flexible structure; ultralow-frequency modal identification; Digital Image Correlation method; Eigensystem Realization Algorithm

## 1. Introduction

With the development of aerospace technology, large-scale flexible structures with ultralow-frequency modal characteristics [1–4], such as solar panels and developable trusses [5], have been widely applied. In order to determine dynamics behavior of the ultralow-frequency flexible structure and ensure the safety in the space environment, it is necessary to conduct modal testing on the structure [6–11].

Modal testing methods are divided into two categories: contact measurement and non-contact measurement. Traditional modal tests generally use contact measurement methods, using sensors, etc. to obtain structural response signals. The main problem of the contact measurement method is that it will bring additional mass and constraints to the lightweight and flexible structure, which will affect its dynamics performance and measurement results [12–14]. The influence of the additional quality of the sensors are always need to eliminate. Cakar et al. [15] proposed a method based on the Sherman–Morrison identity to eliminate the additional mass of the sensor. This method uses a virtual mass to eliminate the influence of the additional mass, which can be well applied to the vibrating test. Zhu [16] and others proposed a one-step elimination method based on the

Sherman–Morrison–Woodbury formula, which directly eliminates the mass load effect of the transducer in the modal test.

The non-contact measurement method has become an important research direction of experimental mechanics, among which the optical detection method is the main one. The non-contact measurement method does not need to attach the sensor to the test piece, fundamentally eliminates the influence of additional mass, and has the advantage of high measurement sensitivity. The main methods of optical measurement methods to measure strain include moiré method [17,18], holographic interferometry [19] and digital image correlation method [20].

Digital Image Correlation method (DIC) is an emerging vibration measurement method in recent years. It has the advantages of not affecting the dynamic performance of the structure under test and saving the cost of modal measurement. It is an ideal method for vibration measurement of flexible structures [21–23]. Many scholars have done a lot of research and application on the DIC method, and have obtained many research results. Ha, N.S. et al. [24] proposed a method to obtain the modal parameters of the artificial wings of the beetle's rear wing by using the discrete cosine transform technique based on the three-dimensional DIC method. The results obtained are in good agreement with the finite element analysis. S. Rizo-Patron [25] and others combined the DIC method with the ITD method for the first time to determine the modal parameters of the helicopter rotor blades under working conditions. For the vibration test of ultralow-frequency flexible structures, Trebuňa et al. [26] used the DIC method to excite the steel fan blades with white noise signals for modal analysis, and used the frequency domain decomposition method to determine the mode from the output power spectral density matrix. Regarding modal parameters; Yang et al. [27] proposed a method that can blindly extract the modal frequency, damping ratio and full-field mode shape from the vibration video image of the structure, which improved the problem of difficulty in arranging speckle patterns or targets in structural modal testing with DIC method.

For the study of spacecraft dynamic parameter identification [28,29], frequency domain methods were mostly used in the early days, such as Frequency Domain Decomposition [30]. However, this method requires inverse Fourier transform in terms of damping identification, which is realized by exponential decay method in the time domain, and the accuracy is not high due to the influence of truncation error. For ultralow-frequency flexible structures, many classical frequency domain identification methods are not applicable, so the time domain identification method has gradually become the main method in the identification of spacecraft dynamics parameters. The Eigensystem Realization Algorithm [31] (ERA) is convenient to determine the modal order and the recognition speed is faster due to the short sampling time, high identification accuracy and strong anti-noise ability. At the same time, it has a strong recognition ability for structures with ultralow-frequency modal characteristics. Pappalardo et al. [32] proposed a system identification method of the linear dynamics model of the multi-body mechanical system based on the ERA. Numerical examples of simple vehicle models are used to verify the effectiveness of the proposed recognition method. Juang [33] proposed an improved method for the ERA, usually called the Eigensystem Realization Algorithm using data correlations (ERA/DC). Compared with the traditional ERA, the modal identification is carried out through the flexible truss structure, and the data correlation is used to reduce the influence of noise in the identification of modal parameters.

This paper proposes an ultralow-frequency flexible structure modal identification method based on the DIC method. Taking the ultralow-frequency flexible structure as the research object, designing the ultralow-frequency flexible structure excitation system, and performing initial displacement excitation on the structure. The response signal of the ultralow-frequency flexible structure is obtained by the DIC method of the camera array, and the modal frequency is identified using the ERA, in which the first modal frequency is the ultralow-frequency less than 0.5 Hz. Establish an ultralow-frequency flexible structure ground test system based on the DIC method. First, carry out the modal

test and identification of the traditional contact measurement and the DIC method of the solar wing array to verify the accuracy of the DIC method. Besides, based on the Initial Displacement Excitation Method and DIC method, the modal identification of the ultralow-frequency flexible structure of the cantilever beam is carried out. The theoretical results are compared to verify the feasibility and accuracy requirements of the ground modal test system for the dynamic characteristics of the flexible structure designed in this paper.

## 2. Basic Theory

### 2.1. Initial Displacement Excitation Method

The forced vibration motion equation of a damped multi-degree of freedom system is

$$M\ddot{x} + C\dot{x} + Kx = F(t) \tag{1}$$

where: $M$ is a mass matrix, which is a positive definite matrix; $C$ is a proportional damping matrix; $K$ is a stiffness matrix, which is a positive definite or semi-positive definite matrix; $F(t)$ is an external force vector. Through generalized eigenvalue analysis, the first $n$-order natural frequencies $(\omega_1, \omega_2, \ldots, \omega_n)$ and natural modes $(\varphi_1, \varphi_2, \ldots, \varphi_n)$ of the multi-degree-of-freedom system are obtained. The $n$-order mode vector can form the following mode matrix $\phi$.

$$\phi = (\varphi_1 \; \varphi_2 \; \cdots \; \varphi_n) \tag{2}$$

According to the orthogonality of modes, any $n$-dimensional vibration of the system can be uniquely expressed as a linear combination of modes.

$$x = \phi z = \sum_{r=1}^{n} \varphi_r z_r \tag{3}$$

Among them, $z_r (r = 1, 2, \ldots, n)$ is the generalized coordinate describing the motion of the system in the modal space, called the principal coordinate, and $r$ represents the modal order. The array $z$ composed of the principal coordinates of each order is the principal coordinate array.

$$z = \begin{pmatrix} z_1 & z_2 & \cdots & z_n \end{pmatrix}^T \tag{4}$$

Substituting the principal coordinates established by the formula into the equation, and multiplying each item to the left by $\boldsymbol{\phi}^{-1}$ can be obtained.

$$M\ddot{z} + C\dot{z} + Kz = 0 \tag{5}$$

Assuming that $C$ is proportional damping, $n$ independent single-degree of freedom systems are obtained after modal decomposition.

$$m\ddot{z}_r + c\dot{z}_r + kz_r = 0 \tag{6}$$

Assuming that the initial condition of the structure is an arbitrary overall initial displacement, namely.

$$t = 0 : x(0) \neq 0, \; \dot{x}(0) = 0 \tag{7}$$

At this time, the solution of Equation (6) is:

$$z_r(t) = \frac{z_r(0)}{\sqrt{1 - \xi_r^2}} e^{-\xi_r \omega_r t} \cos(\omega_{dr} t - \theta_r) \tag{8}$$

Among them, $\omega_r$ is the undamped natural frequency, $\omega_{dr}$ is the natural frequency of the damping system, and $\xi_r$ is the damping ratio.

The initial displacement $z_r(0)$ of the modal space can be expressed as:

$$z_r(0) = \frac{\varphi_r^T M x(0)}{\varphi_r^T M \phi_r} \tag{9}$$

Incorporate formula (8) into formula (3) to get the system response.

$$x(t) = \sum_{r=1}^{n} \varphi_r \frac{z_r(0)}{\sqrt{1-\xi_r^2}} e^{-\xi_r \omega_r t} \cos(\omega_{dr} t - \theta_r) \tag{10}$$

In formula (10), $\cos(\omega_{dr} t - \theta_r)$ is expressed as:

$$\cos(\omega_{dr} t - \theta_r) = \frac{e^{i(\omega_{dr} t - \theta_r)} + e^{-i(\omega_{dr} t - \theta_r)}}{2} \tag{11}$$

Then the system response can be expressed as:

$$x(t) = \sum_{r=1}^{n} \frac{1}{2} \frac{\varphi_r z_r(0)}{\sqrt{1-\xi_r^2}} e^{(-\xi_r \omega_r + i\omega_{dr})t} e^{-i\theta_r} + \sum_{r=1}^{n} \frac{1}{2} \frac{\varphi_r z_r(0)}{\sqrt{1-\xi_r^2}} e^{(-\xi_r \omega_r - i\omega_{dr})t} e^{i\theta_r} \tag{12}$$

Under any integral initial displacement condition, the system response has the form of impulse response function, and the impulse response function is the input condition of the characteristic system to realize the algorithm.

### 2.2. Digital Image Correlation Method

The Digital Image Correlation method is based on the binocular vision theory. Its basic idea is to use two cameras to shoot the same target, and use the straightness and intersection of the left and right cameras to perform inverse solution to obtain the 3D space coordinate value of the target. As shown in Figure 1, the three-dimensional space target point $P(x_w, y_w, z_w)$ respectively forms the point $P_1(u_1, v_1)$ and the point $P_2(u_2, v_2)$ in the image coordinate system of the left and right cameras. The origins $O_{c1}$ and $O_{c2}$ are connected to the point $P$ in the three-dimensional space, and the point $P$ is located on the two imaging rays of the left and right cameras at the same time. The imaging model of the camera can be reversely solved.

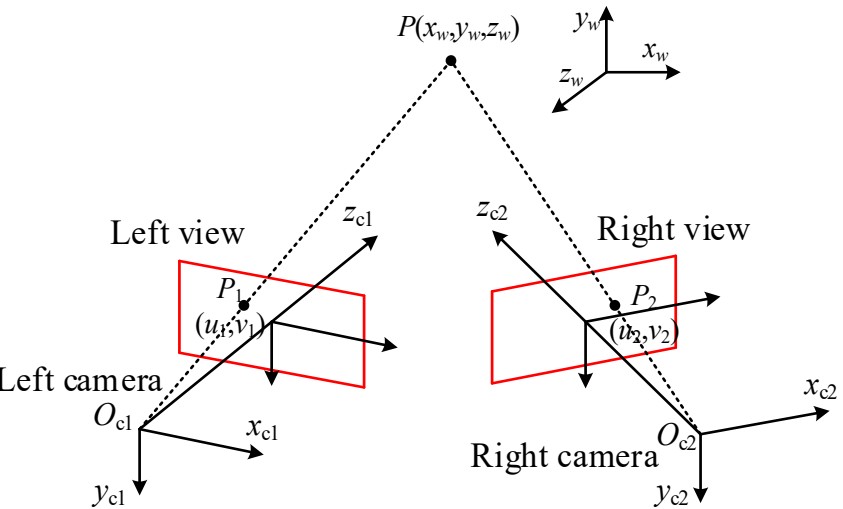

**Figure 1.** Principle of three-dimensional coordinate reconstruction algorithm.

Through camera calibration, the projection matrices *M1* and *M2* of the left and right cameras can be obtained. The projection matrix of the camera combines the target point $P(x_w, y_w, z_w)$ in the three-dimensional space with the imaging points $P_1(u_1, v_1)$ and $P_2(u_2,$

$v_2$) in the camera image coordinate system. See formulas (13) and (14) for details. $Z_{c1}$ and $Z_{c2}$ are the magnification coefficients in the camera imaging model and are related to the internal parameters of the camera.

$$z_{c_1} \begin{bmatrix} u_1 \\ v_1 \\ 1 \end{bmatrix} = M1 \begin{bmatrix} x_w \\ y_w \\ z_w \\ 1 \end{bmatrix} = \begin{bmatrix} m_{11}^1 & m_{12}^1 & m_{13}^1 & m_{14}^1 \\ m_{21}^1 & m_{22}^1 & m_{23}^1 & m_{24}^1 \\ m_{31}^1 & m_{32}^1 & m_{33}^1 & m_{34}^1 \end{bmatrix} \begin{bmatrix} x_w \\ y_w \\ z_w \\ 1 \end{bmatrix} \tag{13}$$

$$z_{c_2} \begin{bmatrix} u_2 \\ v_2 \\ 1 \end{bmatrix} = M2 \begin{bmatrix} x_w \\ y_w \\ z_w \\ 1 \end{bmatrix} = \begin{bmatrix} m_{11}^2 & m_{12}^2 & m_{13}^2 & m_{14}^2 \\ m_{21}^2 & m_{22}^2 & m_{23}^2 & m_{24}^2 \\ m_{31}^2 & m_{32}^2 & m_{33}^2 & m_{34}^2 \end{bmatrix} \begin{bmatrix} x_w \\ y_w \\ z_w \\ 1 \end{bmatrix} \tag{14}$$

Simultaneous formulas (13) and (14), after eliminating the amplification factor $Z$, four equations are obtained, as shown in formula (15), in which there are three unknowns ($x_w$, $y_w$, $z_w$), and the least square method is used to solve the statically indeterminate equations to obtain the three-dimensional space coordinate values ($x_w$, $y_w$, $z_w$) of the target point $P$.

$$\begin{cases} (u_1 m_{31}^1 - m_{11}^1) x_w + (u_1 m_{32}^1 - m_{12}^1) y_w + (u_1 m_{33}^1 - m_{13}^1) z_w = m_{14}^1 - u_1 m_{34}^1 \\ (v_1 m_{31}^1 - m_{21}^1) x_w + (v_1 m_{32}^1 - m_{22}^1) y_w + (v_1 m_{33}^1 - m_{23}^1) z_w = m_{14}^1 - v_1 m_{34}^1 \\ (u_2 m_{31}^2 - m_{11}^2) x_w + (u_2 m_{32}^2 - m_{12}^2) y_w + (u_2 m_{33}^2 - m_{13}^2) z_w = m_{14}^2 - u_2 m_{34}^2 \\ (v_2 m_{31}^2 - m_{21}^2) x_w + (v_2 m_{32}^2 - m_{22}^2) y_w + (v_2 m_{33}^2 - m_{23}^2) z_w = m_{14}^2 - v_2 m_{34}^2 \end{cases} \tag{15}$$

As shown in Figure 2, select the square image sub-area to be calculated in the digital image taken by the left camera before deformation. And find the corresponding position in the digital image taken by the right camera before deformation according to the correlation matching between the left and right cameras. According to the pre-calibrated internal and external parameters of the camera, the three-dimensional coordinates ($x_0$, $y_0$, $z_0$) of the center point of the image subregion. In the same way, taking the picture taken by the left camera before the deformation as the reference picture, the digital images collected by the left and right cameras after the deformation are also accurately tracked to the corresponding position of the calculation area, and the space three-dimensional coordinates of the deformed point can also be obtained ($x_1$, $y_1$, $z_1$), the difference between the space coordinates before and after the deformation is the three-dimensional displacement ($x$, $y$, $z$) of this point.

In engineering applications, the measured object may undergo major deformation or rotation, and a square reference subregion may no longer be square after deformation. Therefore, the shape function is introduced to correspond the pixels in the reference subregion and the target subregion. The first order shape function allows translation and rotation of the target subregion, as well as uniform shearing and stretching deformation, which is suitable for most situations. Its expression is:

$$w(\xi, p) = \begin{bmatrix} 1 + u_x & u_y & u \\ v_x & 1 + v_y & v \\ 0 & 0 & 1 \end{bmatrix} \begin{bmatrix} \Delta x \\ \Delta y \\ 1 \end{bmatrix} \tag{16}$$

The incremental function $w = (\xi, \Delta p)$ of the shape function can be expressed as:

$$w(\xi, \Delta p) = \begin{bmatrix} 1 + \Delta u_x & \Delta u_y & \Delta u \\ \Delta v_x & 1 + \Delta v_y & \Delta v \\ 0 & 0 & 1 \end{bmatrix} \begin{bmatrix} \Delta x \\ \Delta y \\ 1 \end{bmatrix} \tag{17}$$

Use the inverse Newton–Gauss iteration method (IC-GN), and its second order shape function is shown in formula (18).

$$x' = x + u + u_x\Delta x + u_y\Delta y + \frac{1}{2}u_{xx}\Delta x^2 + \frac{1}{2}u_{yy}\Delta y^2 + u_{xy}\Delta x\Delta y$$
$$y' = y + v + v_x\Delta x + v_y\Delta y + \frac{1}{2}v_{xx}\Delta x^2 + \frac{1}{2}v_{yy}\Delta y^2 + v_{xy}\Delta x\Delta y$$

(18)

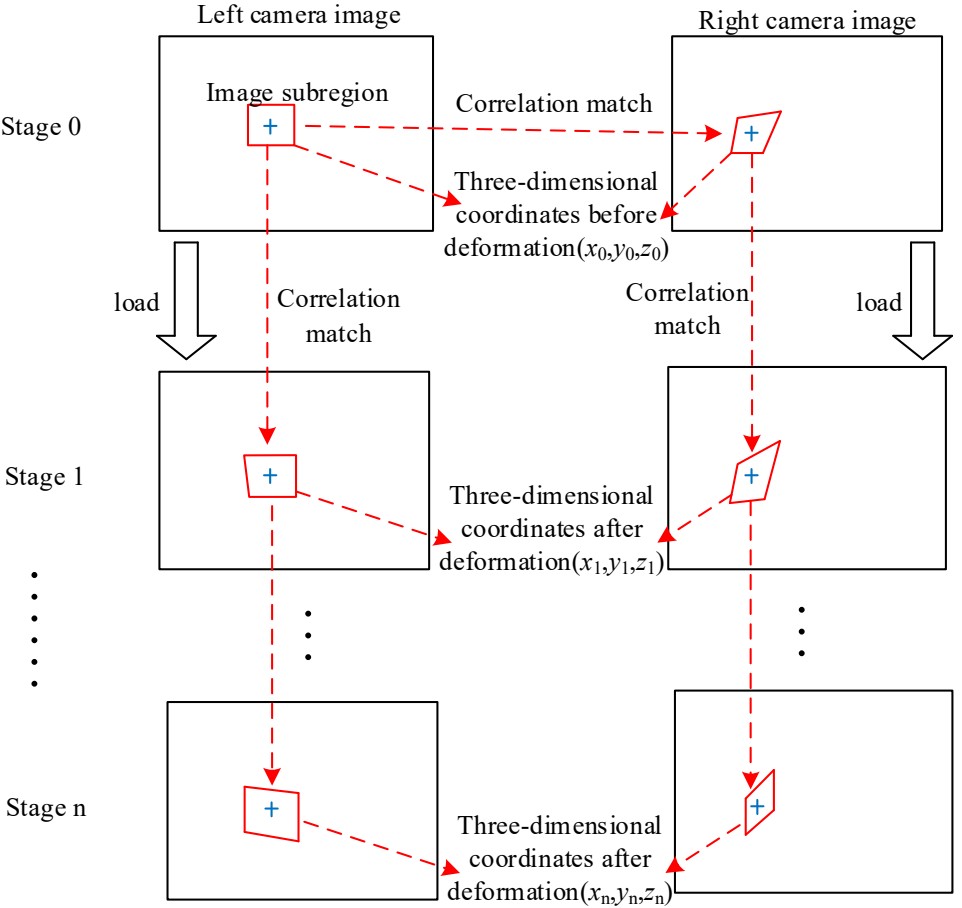

**Figure 2.** Calculation steps of 3D displacement.

The correlation function is a function that measures the similarity between the reference subregion and the target subregion. The selection of the correlation function directly affects the final accuracy of the calculation. When the function takes the minimum value of 0, it can be considered that there is no difference between the reference subregion and the target subregion at this time, that is, the best match. However, even under the most ideal test conditions, the digital images of stationary objects taken at different times are still different. The reasons can be attributed to camera noise, illumination changes, and surface changes of the specimen due to deformation, and so on. Therefore, the correlation function should have a certain degree of stability in addition to quantifying the difference between the subregions. The zero-mean normalized least square distance correlation function can effectively avoid the mismatch phenomenon caused by the change of light intensity, and ensure the stability of the experiment. Its expression is:

$$C_{ZNSSD} = \sum_{\zeta}\left\{\frac{\overline{f}[\psi + w(\xi,\Delta p)]}{\sqrt{\sum_{\varsigma}\left\{\overline{f}[\psi + w(\xi,\Delta p)]\right\}^2}} - \frac{\overline{g}[\psi + w(\xi,p)]}{\sqrt{\sum_{\varsigma}\left\{\overline{g}[\psi + w(\xi,\Delta p)]\right\}^2}}\right\}^2$$

(19)

where $C_{ZNSSD}$ is the optimization coefficient, $p = [u, u_x, u_y, v, v_x, v_y]$ is the deformation parameter vector, and $\Delta p = [\Delta u, \Delta u_x, \Delta u_y, \Delta v, \Delta v_x, \Delta v_y]$ is its incremental vector, $\psi = [x, y, 1]$ represents the whole pixel coordinates of the selected test point in the subregion of the image, $\xi = [\Delta x, \Delta y, 1]^T$ represents the sub-pixel coordinates corresponding to the whole pixel coordinate point.

Perform a firstorder $\Delta p$ expansion on $C_{ZNSSD}(\Delta p)$, and get:

$$
C_{ZNSSD}(\Delta p) = \sum_{\zeta} \left\{ \frac{\overline{f}(\psi + \xi) + \nabla f(\psi + \xi)\dfrac{\partial w}{\partial p}}{\sqrt{\sum_{\varsigma}\left\{\overline{f}[\psi + w(\xi, \Delta p)]\right\}^2}} - \frac{\overline{g}[\psi + w(\xi, p)]}{\sqrt{\sum_{\varsigma}\{\overline{g}[\psi + w(\xi, \Delta p)]\}^2}} \right\}^2 \tag{20}
$$

where $\nabla f(\psi + \xi) = [\partial f(\psi + \xi)/\partial x, \partial f(\psi + \xi)/\partial y]$ represents the gray gradient of the reference image subregion, and the Jacobian matrix of the shape function is expressed as $\dfrac{\partial w}{\partial p} = \begin{bmatrix} 1 & \Delta x & \Delta y & 0 & 0 & 0 \\ 0 & 0 & 0 & 1 & \Delta x & \Delta y \end{bmatrix}$. When the reference image subarea is most similar to the target subarea, $C_{ZNSSD}(\Delta p)$ gets the minimum value, which can be obtained by $\partial C_{ZNSSD}(\Delta p)/\partial \Delta p = 0$, where $H^{-1}$ is the inverse of the Hessain matrix.

$$
\begin{aligned}
\Delta p &= H^{-1}\left\{ \left[\nabla f(\psi + \xi)\frac{\partial w}{\partial p}\right]^T \left[\frac{\overline{f}_n}{\overline{g}_n}\overline{g}\left[\psi + w(\xi, p) - \overline{f}(\psi + \xi)\right]\right] \right\} \\
H &= \sum_{\varsigma}\left\{ \left[\nabla f(\psi + \xi)\frac{\partial w}{\partial p}\right]^T \left[\nabla f(\psi + \xi)\frac{\partial w}{\partial p}\right] \right\}
\end{aligned} \tag{21}
$$

*2.3. Eigensystem Realization Algorithm*

The input condition of the ERA is the impulse response function. By solving the response of the damped multi-degree of freedom vibration system under arbitrary displacement excitation, it is found that the response solution under the initial displacement excitation can replace the impulse response function as the input of the ERA.

For a finite, discrete-time linear time-invariant system, the state equation can be expressed in the following form:

$$
\begin{cases} x(k+1) = Ax(k) + Bu(k) \\ \quad\quad y(k) = Cx(k) \end{cases} \tag{22}
$$

Among them, $x \in R^n, u \in R^m, y \in R^p$ are the state vector, input vector and output vector, respectively, and $A$, $B$, $C$ are the state matrix, input matrix and output matrix of the system, respectively.

The Hankel matrix is constructed using the response function under arbitrary initial displacement excitation, namely:

$$
H_{rs}(k-1) = \begin{bmatrix} Y(k) & Y(k+t_1) & \cdots & Y(k+t_{s-1}) \\ Y(j_1+k) & Y(j_1+k+t_1) & \cdots & Y(j_1+k+t_{s-1}) \\ \vdots & \vdots & & \vdots \\ Y(j_{r-1}+k) & Y(j_{r-1}+k+t_1) & \cdots & Y(j_{r-1}+k+t_{s-1}) \end{bmatrix} \tag{23}
$$

where $Y(k) \in R^{l \times p}$ is the response function matrix under arbitrary initial displacement excitation, namely.

$$Y(k) = \begin{bmatrix} h_{11}(k) & h_{12}(k) & \cdots & h_{1p}(k) \\ h_{21}(k) & h_{22}(k) & \cdots & h_{2p}(k) \\ \vdots & \vdots & & \vdots \\ h_{L1}(k) & h_{L2}(k) & \cdots & h_{Lp}(k) \end{bmatrix}_{(l \times p)} \tag{24}$$

In the formula, $h_{ij}(k)$ is the displacement excitation response function between excitation point $j$ and response point $i$ at time $k$. Perform singular value decomposition on $H_{rs}(0) = UVW^T$; the order determined by singular value decomposition obtains the minimum realization of the system:

$$A = V^{-\frac{1}{2}} U^T H_{rs}(1) W V^{-\frac{1}{2}} \tag{25}$$

$$B = V^{\frac{1}{2}} W^T E_m \tag{26}$$

$$C = E_P^T U V^{\frac{1}{2}} \tag{27}$$

where: $E_p{}^T = [I_p, 0_p, \dots, 0_p]$, $E_m{}^T = [I_m, 0_m, \dots, 0_m]$.

Carry out the eigenvalue decomposition of the matrix $A$ to obtain the eigenvalue matrix $G$, and then obtain the eigenvector matrix $\varphi$.

$$\varphi^{-1} A \varphi = G, G = diag(g_1, g_2, \cdots, g_r) \tag{28}$$

where: $g_r$ is the eigenvalue of matrix $A$, and $r$ is the modal order.

Determine the modal frequency $\omega_r$ and modal damping $\zeta_r$ according to the relationship between the eigenvalue gr of the matrix $A$ and the system eigenvalue $\lambda_r$:

$$\lambda_i = \frac{1}{\Delta t} \ln g_i = \lambda_r{}^R + j \lambda_r{}^I \tag{29}$$

$$\omega_r = \sqrt{(\lambda_r{}^R)^2 + (\lambda_r{}^I)^2} \tag{30}$$

$$\zeta_r = -\frac{\lambda_r^R}{\sqrt{(\lambda_r^R)^2 + (\lambda_r^I)^2}} \tag{31}$$

In the formula: $\lambda_r{}^R$ represents the real part of the system eigenvalue $\lambda_r$, $\lambda_r{}^I$ represents the imaginary part, and $r$ is the modal order.

According to the output matrix $C$ and the eigenvector matrix $\varphi$, the mode shape matrix $\phi$ can be determined:

$$\phi = C\varphi = E_P^T U V^{\frac{1}{2}} \varphi \tag{32}$$

Use the Modal Assurance Criterion (MAC) to check the independence and consistency between the two modes:

$$MAC_{uv} = \frac{\left| \phi_u{}^T \phi_v \right|^2}{\phi_u{}^T \phi_u \phi_v{}^T \phi_v} \tag{33}$$

Among them, $\phi_u$ and $\phi_v$ are mode vectors, which are column vectors, and $u$ and $v$ are modal orders. In the formula, modal displacement needs to be normalized, and its value ranges from [0, 1]. The MAC value of 1 indicates that the two modes are completely correlated, while a value of 0 indicates that the two modes are completely unrelated.

The modal identification process of ultralow-frequency flexible structure is shown in Figure 3:

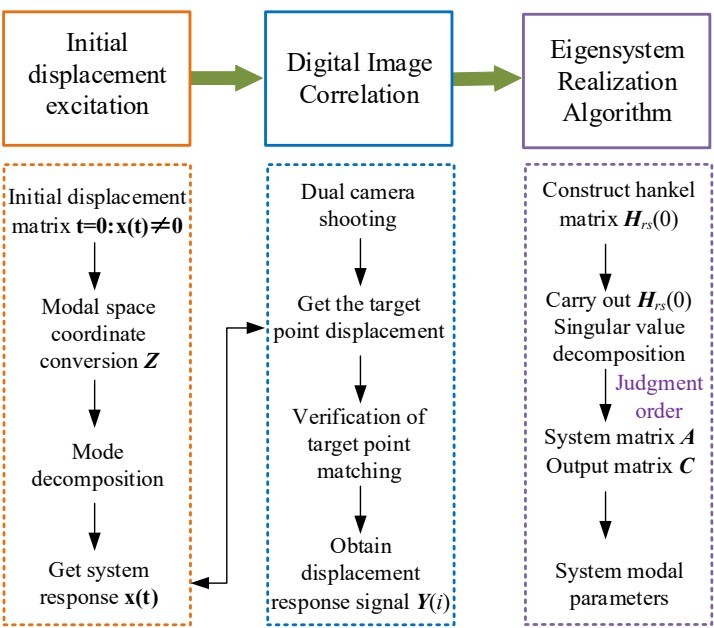

**Figure 3.** Flow chart of modal identification of ultralow-frequency flexible structure.

## 3. Case Study

### 3.1. Modal Test Verification of Low Frequency Flexible Solar Wing

In order to preliminarily verify the proposed ultralow-frequency flexible structure modal test method based on the Digital Image Correlation method, the ultralow-frequency flexible solar wing array in the unfolded state is taken as the test object. The flexible solar wing is composed of two solar wings with exactly the same size. The material properties are shown in Table 1. The flexible solar wing array is suspended by a suspension rope, and its size and installation are shown in Figure 4:

**Table 1.** Solar wing material properties.

| Density/(kg/m³) | Elastic Modulus/(GPa) | Poisson's Ratio | Thickness/(mm) |
| --- | --- | --- | --- |
| 2700 | 70 | 0.33 | 5 |

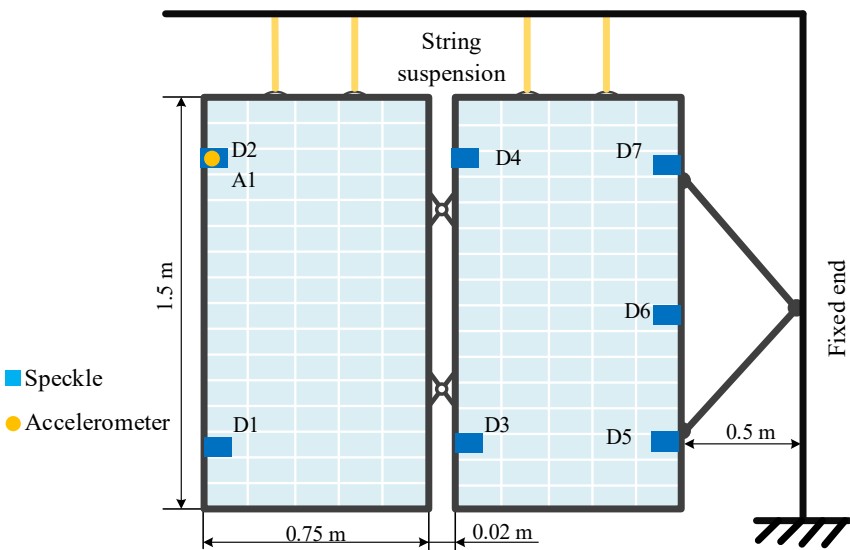

**Figure 4.** Dimensional drawing of ultralow-frequency flexible solar wing array with the speckle pasting position and the camera placement position.

The equipment used in the test is as follows:

1. DHDAS dynamic signal acquisition and analysis system, the sampling frequency is 25.6 Hz, the sampling point is 512, and the frequency domain resolution is 0.05 Hz;
2. A piezoelectric acceleration sensor, model CA-YD-107 sensitivity 2.73 pC/ms$^{-2}$;
3. Force hammer, model CL-YD-303, sensitivity 3.99 pC/N;
4. Two SONY FDR-AX40 cameras, the camera frame number is 20 fps, that is, the sampling frequency is 20 Hz;

Paste the speckle patches on the side of the solar wing mechanism, a total of 7, numbered from D1 to D7, and the acceleration sensor A1 is arranged at the spot of D2 which are shown in Figure 4.

The test adopts multi-point excitation and single-point measurement, and is carried out by the hammering method. Seven points such as D1–D7 are selected as excitation points, and each measurement point is hammered in turn on the back of the solar wing mechanism, and the structure response is recorded by the DHDAS dynamic signal acquisition and analysis system. At the same time, the image of the solar wing mechanism is collected through the camera, and the displacement response data of the measuring point in 0–20 s is calculated by the DIC method, and the modal parameters of the solar wing mechanism are obtained through modal identification. The traditional contact measurement results are compared with the DIC test results to verify the feasibility and accuracy of the proposed ultralow-frequency flexible structure modal test based on the Digital Image Correlation method.

Figure 5 is the amplitude-frequency diagram of DHDAS and DIC method. Table 2 shows the comparison between DHDAS system identification frequency and DIC identification frequency. From the comparison results, it can be seen that the DHDAS system identification frequency and the DIC method identification frequency error are small, the first order frequency error is 2.80%, and the second order frequency error is −1.30%, indicating that the effective optical test data can be used to obtain a higher-precision structure. The modal frequency results verify the feasibility and accuracy of the proposed ultralow-frequency flexible structure modal test method based on the DIC method.

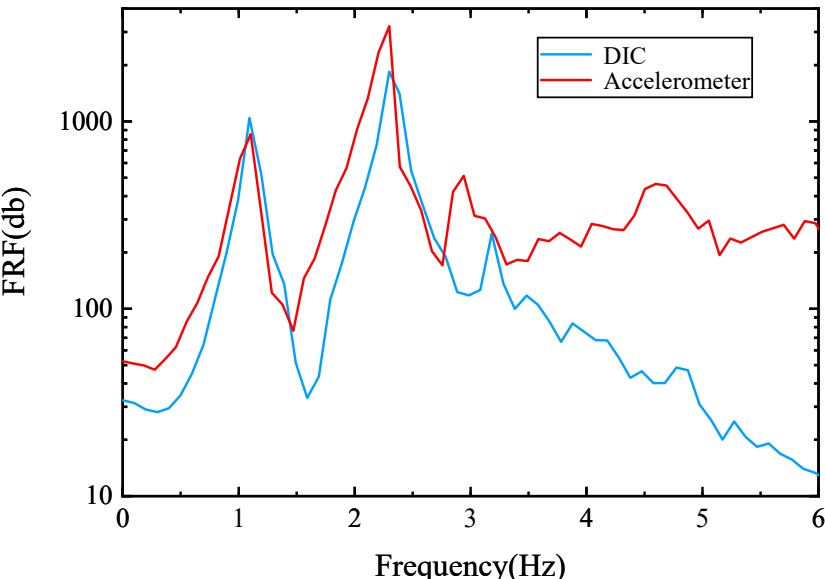

**Figure 5.** Frequency response function diagram.

**Table 2.** Comparison of modal parameter recognition results.

| Modal Order | DHDAS Identification Frequency/Hz | DIC Method Identification Frequency/Hz | Frequency Error/% |
|---|---|---|---|
| First order (Bending) | 1.07 | 1.10 | 2.80 |
| Second order (Twisting) | 2.29 | 2.26 | −1.30 |

The results of the DHDAS system's identification and the DIC method of identification of the vibration shape are shown in Figure 6 below:

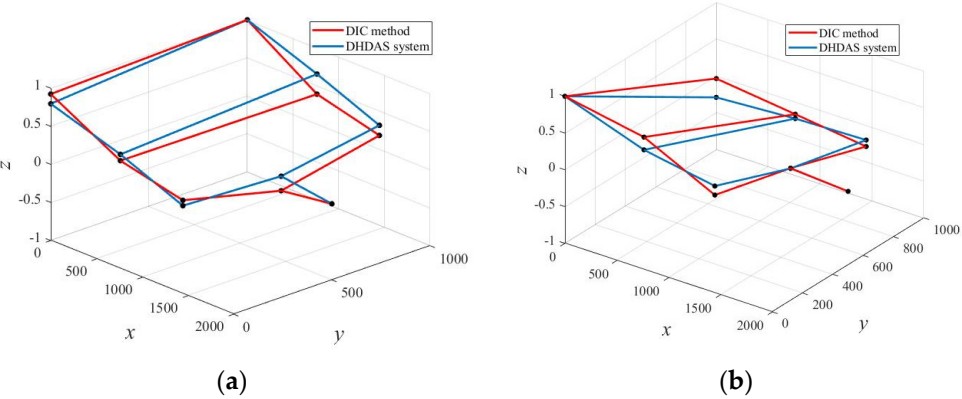

|              (**a**)              |              (**b**)              |

**Figure 6.** Mode shape diagram. (**a**) The first order mode shape. (**b**) The second order mode shape.

According to the theory of modal confidence, the MAC value of mode shape of DHDAS identification is compared with that of the mode shape of DIC method identification. It can be seen from Figure 7 that the modal shape obtained based on the DIC method is compared with the modal shape obtained by the DHDAS system. The matching degree is high, and the MAC value contrast is greater than 0.9, which verifies the feasibility of the low-frequency flexible structure experimental modal test method based on the DIC method.

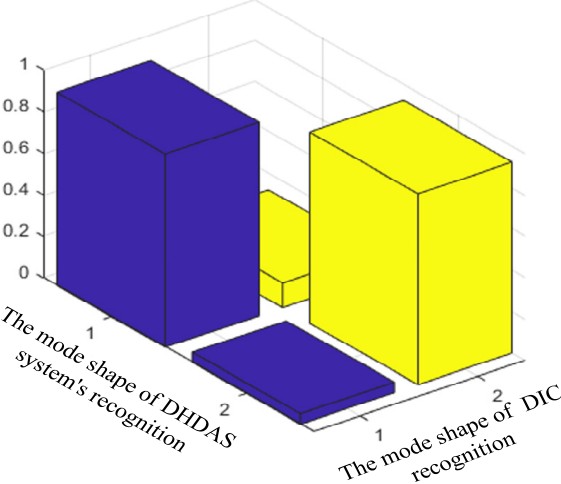

**Figure 7.** The comparison of the MAC value of the mode shape of DHDAS system's identification and the mode shape of DIC identification.

### 3.2. Structural Modal Identification of Low Frequency Flexible Suspension Beam

For flexible beams with concentrated masses, based on the designed ultralow-frequency flexible structure dynamic characteristics ground test system, the ultralow-frequency flexible structure dynamic characteristics ground test research is carried out, the modal parameters of the structure are obtained, and the modal identification results are compared with

the finite element theory results. Contrast, verify the feasibility and accuracy requirements of the ultralow-frequency flexible structure dynamic characteristics ground test system designed in this paper.

The test object is composed of two horizontally suspended slender flexible beams spliced by a connecting plate. The total length is 3 m, and it is divided into 16 measurements. Fix concentrated masses at nodes 3~7 and nodes 10~14, as shown in Figure 8.

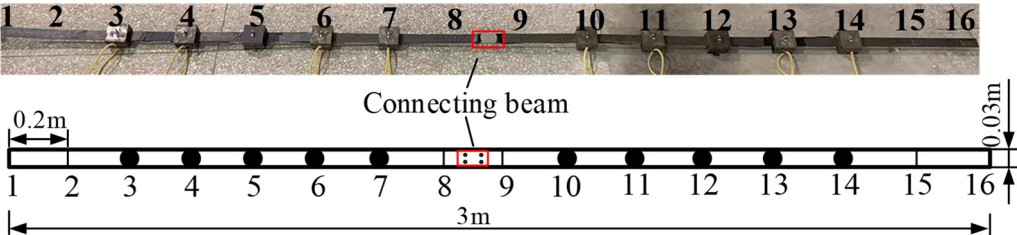

**Figure 8.** Dimension drawing of flexible beam.

The material properties of the suspended fixed flexible beam are shown in Table 3:

**Table 3.** Material attribute table of suspended fixed flexible beam.

| Material Properties | Numerical Value |
|---|---|
| Elastic Modulus/(Pa) | $2.1 \times 10^{11}$ |
| Density/(kg/m$^3$) | $7.85 \times 10^3$ |
| Poisson's ratio | 0.34 |
| Section size/(m) | $0.03 \times 0.002$ |
| Length/(m) | 3 |

A concentrated mass block is installed on both sides of each node of the flexible beam and fixed to the flexible beam through threaded holes. The eye hole is used for suspension and the initial displacement application hole is used for fixing. The initial displacement is applied to the flexible beam during the test.

The specific dimensions and material properties of the concentrated mass are shown in Table 4:

**Table 4.** Concentrated mass size and material attribute table.

| Material Properties | Numerical Value |
|---|---|
| Elastic Modulus/(Pa) | $2.1 \times 10^{11}$ |
| Density (kg/m$^3$) | $7.85 \times 10^3$ |
| Poisson's ratio | 0.34 |
| Size/(m) | $0.06 \times 0.06 \times 0.002$ |
| Mass/(kg) | 0.565 |

The target is the target taken by the camera, and each target is a group of speckles. The speckle detail diagram is shown in the Figure 9. The speckle pattern is pasted onto a concentrated mass block to form a target.

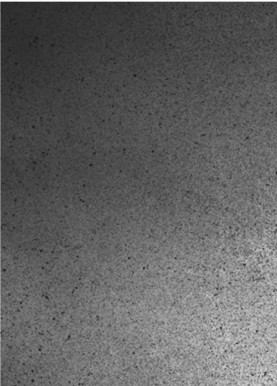

**Figure 9.** Speckle detail diagram.

The test equipment mainly includes:

1.  DHDAS dynamic signal acquisition and analysis system, the sampling frequency is 25.6 Hz, the sampling point is 512, and the frequency domain resolution is 0.05 Hz;
2.  Piezoelectric acceleration sensor, the model is CA-YD-107, the sensitivity is 2.73 pC/ms$^{-2}$;
3.  3 XDA-40/25 electromagnets and 1 24 V student power supply;

The 3 m-long flexible beam is suspended using four-stage flying beams, and the flexible beam nodes 3, 4, 6, 7, 10, 11, 13, and 14 are suspended using rubber ropes, and the flexible beams are suspended at nodes 4, 6, 8, and 10. A total of six targets are arranged in 12 and 14, the node 1 is fixed by the fixing system, the initial displacement of the node 16 is respectively applied by the ultralow-frequency flexible structure excitation system, and they are released synchronously, and the appropriate DIC method is selected to measure the distance and pixel resolution, Among them, each node shooting camera is shown in Table 5. Collect the displacement signal of the structure within 0~100 s, the test site is shown in Figure 10, and the schematic diagram of the test layout is shown in Figure 11.

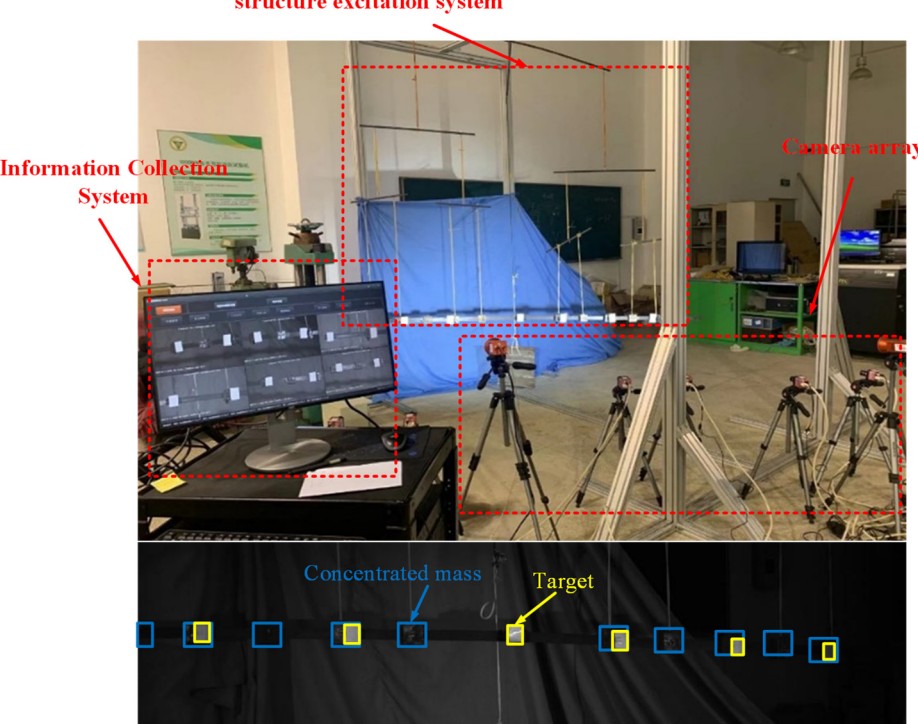

**Figure 10.** Test site map.

**Table 5.** Camera settings.

| Target (Node) | 1 (4) | 2 (6) | 3 (8) | 4 (10) | 5 (12) | 6 (14) |
|---|---|---|---|---|---|---|
| Shooting camera serial number | 1, 2 | 1, 2 | 3, 4 | 3, 4 | 5, 6 | 5, 6 |

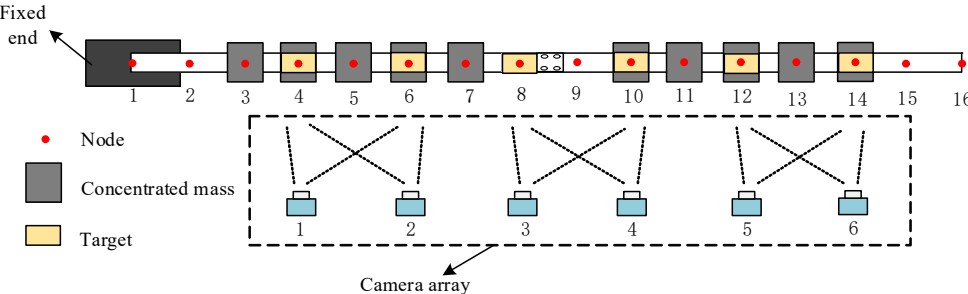

**Figure 11.** Schematic diagram of test layout.

The ERA is used to analyze the test data, and the first five order peaks are selected for analysis and calculation, and the first five order modal frequencies of the structure are obtained. Figure 12 shows the displacement response diagram obtained by the DIC.

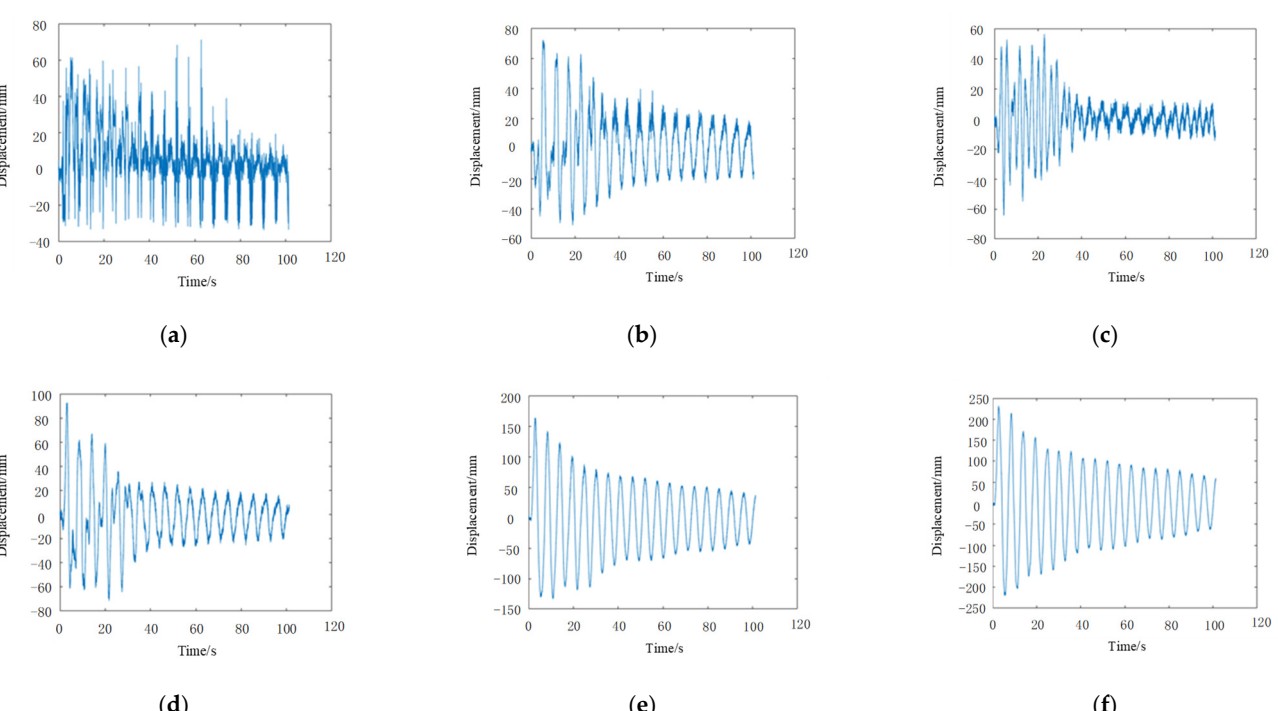

**Figure 12.** Displacement response diagram. (**a**) Displacement response curve of point 1. (**b**) Displacement response curve of point 2. (**c**) Displacement response curve of point 3. (**d**) Displacement response curve of point 4. (**e**) Displacement response curve of point 5. (**f**) Displacement response curve of point 6.

Figure 13 is the test versus theoretical frequency response diagram. Table 6 is the frequency error table based on the ground test system identification based on the dynamic characteristics of the flexible structure and the theory modal frequency. The analysis shows that the error between the first five-order modal parameters identified and the finite element theory result is small, the error of the first order frequency is −1.081%, the error of the second order frequency is 2.661%, the error of the third order frequency is −3.396%, and the fourth order frequency. The error is 1.221%, and the fifth order frequency error is 4.550%,

thus verifying the feasibility and accuracy requirements of the designed flexible structure dynamic characteristics ground test system. The comparison results of the identified mode shape and the simulated mode shape are shown in Table 7:

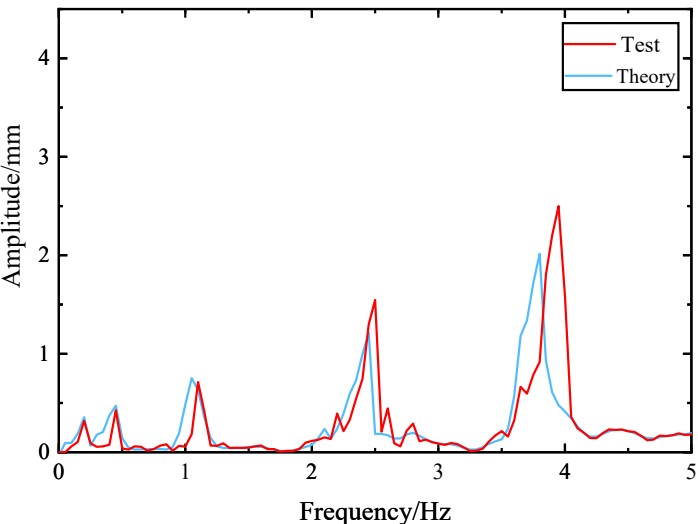

**Figure 13.** Test versus theoretical frequency response diagram.

**Table 6.** Comparison of identification frequency.

| Modal Order | Identification Frequency/Hz | Theory Frequency/Hz | Error/% |
|:---:|:---:|:---:|:---:|
| 1 | 0.183 | 0.185 | −1.081 |
| 2 | 0.463 | 0.451 | 2.661 |
| 3 | 1.081 | 1.119 | −3.396 |
| 4 | 2.488 | 2.458 | 1.221 |
| 5 | 3.975 | 3.802 | 4.550 |

**Table 7.** Comparison diagram of identification modes.

| Modal Order | 1 | 2 | 3 | 4 | 5 |
|:---:|:---:|:---:|:---:|:---:|:---:|
| Identification shape | | | | | |
| Theory shape | | | | | |

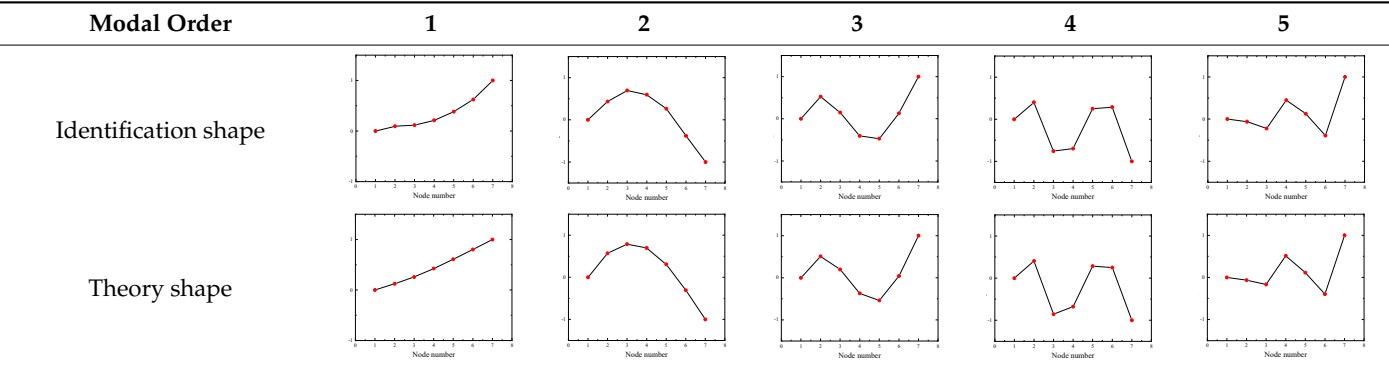

According to the modal confidence theory, the MAC value of the result obtained by the ERA and the mode shape result obtained by Patran is compared, as shown in Figure 14. The first five-order experimental modal vibration obtained by the ERA after the random initial displacement is applied to the structure. The model has a high degree of matching with the modal shape obtained by theory, and the modal confidence can reach more than 0.9, which verifies the feasibility and accuracy requirements of the ground modal test system for the dynamic characteristics of the flexible structure designed in this paper.

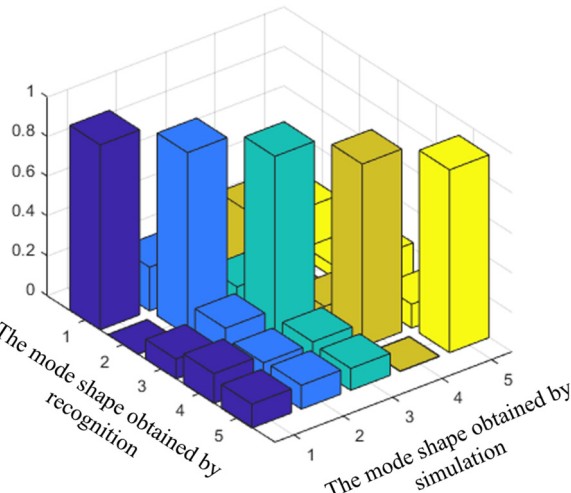

**Figure 14.** Comparison of the MAC value of the identified mode shape and the theoretical mode shape.

## 4. Conclusions

This paper proposes a systematic modal identification method for ultralow-frequency flexible structures. Through the initial displacement excitation of the ultralow-frequency flexible structure, the DIC method is used to obtain the response signal of the structure, and the ERA is used to identify the ultralow-frequency mode of the structure. Through the study of two examples, the following conclusions are obtained:

1.  For the modal test of the solar wing array structure, the DIC method and the traditional contact measurement method were used to identify the modal parameters. The structure frequency identification error of the two modal identification methods is below ±3%, which preliminarily verifies the feasibility and accuracy of the ultralow-frequency flexible structure modal test method based on the DIC method.

2.  Aiming at the ultralow-frequency flexible cantilever beam structure, the modal measurement is carried out by using the DIC method. The experimental test results are in good agreement with the theory results, which proves the accuracy of the modal test method based on the DIC method proposed in this paper.

Flexible structures have the characteristics of ultra-low frequency and the first mode frequency can be less than 0.5 Hz. The novelty of this paper is that the low frequency of the structure is difficult to be excited by the pulse excitation obtained by using a force hammer. A frequency below 0.5 Hz can be excited by using the Initial Displacement Excitation Method. It is difficult to identify modal frequencies below 0.5 Hz using accelerometer measurements. Besides, the contact sensor will cause additional mass to the structure, which will greatly affect the modal analysis results of flexible structures. The Digital Image Correlation method can be used to measure ultra-low frequency without additional mass. This paper proposes a modal identification method based on Digital Image Correlation method for modal analysis of large ultra-low frequency flexible structures. The proposed method can identify flexible structures with a modal frequency of 0.183 Hz.

**Author Contributions:** Conceptualization, Y.W.; methodology, Y.W. and H.Q.; software, H.Q.; validation, H.Q. and R.Z.; formal analysis, H.Q.; investigation, H.Q.; resources, D.J.; data curation, Y.W.; writing—original draft preparation, H.Q.; writing—review and editing, H.Q. and Y.W.; visualization, D.Z.; supervision, R.Z.; project administration, D.J.; funding acquisition, D.J. All authors have read and agreed to the published version of the manuscript.

**Funding:** This research was funded by National Natural Science Foundation of China, grant number No. 11602112. This research was funded by The Natural Science Foundation of the Jiangsu Higher Education Institutions of China, grant number 20KJB460003. This research was funded by Qing Lan Project.

**Institutional Review Board Statement:** Not applicable.

**Informed Consent Statement:** Not applicable.

**Data Availability Statement:** Not applicable.

**Acknowledgments:** The authors are grateful for the support from the National Natural Science Foundation of China (No. 11602112), The Natural Science Foundation of the Jiangsu Higher Education Institutions of China (20KJB460003) and the Qing Lan Project.

**Conflicts of Interest:** The authors declare that there is no conflict of interest regarding the publication of this paper.

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
