# Peer review of "Modal Identification of Ultralow-Frequency Flexible Structures Based on Digital Image Correlation Method"

_applsci, doi:10.3390/app12010185_

Round 1

Reviewer 1 Report

The article sounds interesting. However, contributions are not pointed at all. Moreover, no other works are cited, “Error! 348 Reference source not found” happens. Tables and figures are badly positioned. The paper cannot be accepted in the present form.

Author Response

Thanks for your comments concerning our manuscript. Those comments are all valuable and very helpful for revising and improving our paper, as well as the important guiding significance to our research. We have studied the comments carefully and made revision which we hope to meet with your approval. The revised portions are marked in highlighted in the revised manuscript. The revisions in the paper and responses to the comments are in the attachment.

Reviewer 2 Report

The paper is well written and would appeal to the readers of the Applied Sciences Journal. 

Author Response

(The authors gave the same response as above.)

Reviewer 3 Report

Comments on the paper proposed by Qian et al: “Modal identification of ultralow-frequency flexible structures using digital image correlation method”.

A very interesting topic about the application of DIC for the evaluation of the vibration mode of ultra-low-frequency flexible structure. However, there are some points that are not very clear 

Here are reported some considerations:

  1. From a general point of view, in the introduction must be contain the novelty of the study. Some study about the application of DIC to ultralow-frequency flexible structures are presents in literature (such as, D Kumar et al 2019 Sci. Technol. 30 045903 - A novel real time DIC-FPGA -based measurement method for dynamic testing of light and flexible structure). What is the novelty of this study with respect to what already present in literature? Specify in the Introduction.
  2. In the last part of the introduction, from line 99 to 102, is specified that the experimental part, in addition to the solar wing, is focused on the cantilever beam. Why this case study has been analyzed? Specify in the introduction.
  3. During the whole paper structure, the references of the figures are missing. Check out in the Latex file.
  4. In line 161, is specified that the matrix M1 and M2 has been obtained from the calibration of the camera. How the camera has been calibrated?
  5. .In line 178-179 the punctuation is missing and the concept is not very clear. In general, check out all the punctuation of the paper.
  6. In line 258, the pedix g_r is missing.
  7. In line 281, is better to replace the word “parameters” with “material properties”.
  8. In Figure 4 are shown the excitation points and the positioning of the accelerometer. How the impact point has been selected? Why the accelerometer is placed in that way?
  9. In table 2 are reported the comparison of the vibration mode. Then, is present the shape diagram, basically the displacement of the wing, and they seems quite different. However, for the comparison in frequency domain, is suggested to add the Bode Diagram for the comparison, or, at least, Amplitude-Frequency diagram.
  10. From line 329 to 331 is specified that the matching degree is high. How the matching between the two methods has been evaluated? Through a correlation coefficient?
  11. Regard the cantilever beam experimental test, is not very clear how the number of concentrated masses has been selected.
  12. Also for the cantilever beam, add the Bode diagram for the frequencies comparison.
  13. Also in line 245, specify how the degree of matching has been evaluated.
  14. The conclusions are too short. Insert in this chapter a mini review of this study and highlight the novelty of the work.

For all the previous reasons, the reviewer recommends minor amendments of paper for publication in Applied Sciences.

Author Response

(The authors gave the same response as above.)

Reviewer 4 Report

The authors investigate the response of a suspended bean onto exitation. The low-frequency content is measured via an accelerometer and visual detection using cameras. Finally the measurment results are compared.

It is challenging to understand, where the new bit of the manuscript starts and ends. (Non of the basic theory is new.) Can you please highlight that in the main text!

Please choose a reasonable representation for table 7. There is no need to make the table that large. There is no information in the ranges 1 to 3 and -1 to -3. You can just cut the images.

There are several "Error! Reference source not found." Please fix them!

Author Response

(The authors gave the same response as above.)

Round 2

Reviewer 1 Report

All comments were addressed.

Reviewer 4 Report

The authors improved the manuscript and answered all my questions. Finally, I recommend publication.